# Is Basic Training in Palliative Care Sufficient to Guarantee the Improvement of Knowledge and Skills in This Area?—A Medical Knowledge Assessment Study

**DOI:** 10.3390/medsci13030167

**Published:** 2025-09-02

**Authors:** Rita Monteiro, Hugo Ribeiro, César Vinicius José, Joana Brandão Silva, Ricardo Marinho, João Rocha Neves, Marília Dourado

**Affiliations:** 1Intensive Care Unit, Local Health Unit Santo António, 4099-003 Porto, Portugal; u15989@chporto.min-saude.pt (R.M.); ricardomarinho.medicina@chporto.min-saude.pt (R.M.); 2Faculty of Medicine, University of Coimbra, 3000-548 Coimbra, Portugal; cesarviniciusjose@gmail.com (C.V.J.); jrbesilva@gmail.com (J.B.S.); mdourado@fmed.uc.pt (M.D.); 3Community Palliative Care Support Team Gaia, Local Health Unit Gaia and Espinho, 4400-129 Vila Nova de Gaia, Portugal; 4Coimbra Institute for Clinical and Biomedical Research, 3000-548 Coimbra, Portugal; 5Faculty of Medicine, University of Porto, 4200-319 Porto, Portugal; joaorochaneves@hotmail.com; 6Abel Salazar School of Medicine and Biomedical Sciences, 4099-003 Porto, Portugal; 7Marco Family Health Unit, Local Health Unit of Tâmega and Sousa, 4630-409 Marco de Canaveses, Portugal; 8RISE-Health, University of Porto, 4200-319 Porto, Portugal

**Keywords:** palliative care, post-graduate education, medical training, knowledge assessment

## Abstract

**Background**: With the growing population requiring palliative care (PC), it is essential to enhance and expand the availability of this care in Portugal. Throughout both pre-graduate and post-graduate training for doctors, there are limited learning opportunities in this area, necessitating further training to improve the knowledge and skills needed to support patients at the end of their lives. Studies conducted on doctors and medical students have demonstrated a clear need for improved training. Additionally, others have noted significant benefits for residents who underwent post-graduate training in this field while caring for patients nearing the end of life. This study aims to evaluate the impact of a basic training program on improving palliative care knowledge among medical doctors. **Methods**: This was a cross-sectional study, carried out by sharing a questionnaire with the participants enrolled in the “Intensive Palliative Medicine Course” organized by a group of doctors in November 2021. The questionnaire was completed by participants at three moments of training (before, immediately after the course, and 6 months after the course), and it contained questions to assess the theoretical knowledge, skills, and abilities obtained. **Results**: In total, 93 out of 204 (45.58%) doctors responded before training, 80 (39.21%) immediately after training, and 36 (17.64%) six months after training. After training, an improvement in knowledge was observed (*p* = 0.014), which did not appear to persist six months after the course (*p* = 0.574). However, six months after training, an increase in doctors’ confidence in referring patients to Palliative Care Teams or Units (*p* = 0.009) and medicating patients who may be in the last months of their lives (*p* = 0.005) was observed. **Conclusions**: The results regarding theoretical medical knowledge did not seem to stabilize over time, so it may be necessary to create more specific training opportunities for the medical profession. An increase in doctors’ self-confidence in referring and medicating patients with palliative care needs was observed, which may be associated with better medical care provided. It is necessary to carry out more studies to evaluate the impact of combined theoretical and practical training on the knowledge and confidence of doctors in PC.

## 1. Introduction

Advancements in technology, health, and medical knowledge have led to significant changes in demographic characteristics, disease types, and patient needs, resulting in an increase in the population requiring palliative care (PC) in recent decades [1,2].

The shortage of qualified human resources with expertise and/or specific training in PC has been noted as a primary barrier to enhancing accessibility and advancing PC [3,4].

PC professionals need to understand effective communication, symptom management, ethical decision making, spiritual and emotional support, and interprofessional teamwork [5].

A study evaluated the knowledge and skills of fifth-year students at the Faculty of Medicine of the University of Coimbra (FMUC) in Portugal, revealing a lack of information on topics related directly or indirectly to PC. It was found that students who took the optional curricular unit “Palliative Care and Pain Therapy” showed greater confidence in communicating “bad news”, had more theoretical knowledge of PC, and felt more capable of dealing with terminally ill patients [6].

Other consulted studies also conclude that there is a need to improve and broaden PC training in medicine courses [7,8]. This is linked to students and resident doctors feeling more confident in their communication skills and the pharmacological treatment of patients with palliative needs compared to colleagues without pre-graduate training in this area [9,10]. A systematic review of 19 studies with 3253 medical students found that most undergraduate palliative care teaching methods significantly improved students’ knowledge. However, none assessed effects on clinical performance or patient outcomes, and no single teaching approach was identified as best [11].

Studies indicate that medical residents with post-graduate training in PC (theoretical or practical) offer a distinct advantage when managing patients at the end of life, across the various fields of palliative medicine [12,13,14].

A study conducted in Sintra (Portugal) assessed the knowledge levels of General and Family Medicine (GFM) doctors in PC through a unique questionnaire. The results indicated a lack of training in this area among family medical doctors [15].

A basic palliative care training program for Vietnamese physicians substantially enhanced self-assessed knowledge, skills, and attitudes, with these gains mainly maintained at 6–18 months after training. The impact of post-graduate training was also shown in another study in Germany [16].

The purpose of PC is to improve quality of life, without interfering with or impacting a patient’s life expectancy [17]. To achieve this, the following four key pillars of PC are considered: symptomatic control, multidisciplinary teamwork, communication, and support for the caregiver [18,19,20].

This work aims to analyze and evaluate the impact of PC training on the knowledge acquired and retained by doctors before, during, and after attending a 35 h online PC course. Additionally, the aim is to assess the impact of training on the quality of care provided.

## 2. Methods

### 2.1. Study Design

A cross-sectional web-based survey was conducted between October 2021 and June 2022, addressed to medical doctors in Portugal.

As context and support for the survey, a narrative literature review was conducted. Controlled vocabulary from MeSH terms was utilized to conduct a structured search in the PubMed and Scopus databases. The review aimed to identify key educational interventions and their effects on palliative care training, focusing on studies relevant to post-graduate medical education in palliative care published between 2012 and 2022.

The questionnaire was developed and validated by a panel of six palliative care experts from FMUC, each contributing 10 questions independently, based on insights from the review’s findings. In the first draft, items deemed redundant or overly specific were removed. The revised version underwent initial validation by 10 medical specialists from five different fields—internal medicine, General and Family Medicine, anesthesiology, vascular surgery, and physical and rehabilitation medicine—whose feedback prompted further adjustments. A second validation round involved another 10 physicians (including 5 residents and 5 specialists). The final version was reevaluated by the original expert panel, which rated each item from 0 (not appropriate) to 5 (entirely appropriate), with all items receiving the maximum score.

### 2.2. Selection of Participants

Medical doctors who participated in the “Intensive Palliative Medicine Course” were considered eligible to join the study as long as they were practicing clinical medicine.

The course was conducted online from 19 to 22 October 2021, totaling 35 h, and was organized by a group of palliative care physicians and professors of the Faculty of Medicine of the University of Coimbra.

The palliative care course covered key topics like epidemiology, core principles, pain management, symptom control via case discussions, communicating bad news, family conferences, and psychosocial support for caregivers and the bereaved.

Recruitment was conducted using the email addresses provided by participants during course registration. At that time, participants had given their prior consent to receive information related to the course via email. An invitation to participate in the study was then sent to all participants, which included a description of the study objectives and a link to the self-administered online questionnaire created using Google Forms. Additionally, during the first session of the course, participants were invited to collaborate by participating in this survey. Participation in the study was entirely voluntary, and informed consent was obtained electronically through a checkbox at the beginning of the questionnaire. No exclusion criteria regarding medical specialty or clinical practice setting were applied.

### 2.3. Data Collection

Data were collected through a self-administered online questionnaire created and distributed using Google Forms (http://forms.google.com/ (accessed on 1 October 2021)). The estimated completion time for the questionnaire was approximately 15 min. It consisted of three sections, totaling 48 questions. The questionnaire consisted of the following three parts: Part 1 focused on the sociodemographic and professional characteristics of the participants, comprising 14 questions; Part 2 addressed the theoretical knowledge acquired by the participants, consisting of 27 statements with Agree/Disagree response options; and Part 3 explored the practical application of the participants’ knowledge and skills, which included 7 questions. A complete version of the questionnaire is available in the Appendix A. The questionnaire was administered at the following three distinct points in time: before the intervention began, immediately after the training was completed, and six months after the training ended. Participation was entirely voluntary, and all responses were collected anonymously. Informed consent was obtained electronically through a required checkbox at the beginning of the questionnaire. Participants who did not provide consent were not allowed to continue, and their data were not recorded.

Survey responses were exported from Google Forms to Microsoft Excel. The dataset is secured with a password and will be stored for academic purposes for a period not exceeding five years.

### 2.4. Data Analysis

Data analysis was conducted using the SPSS software, version 26.0. For presenting descriptive statistics, means (Ms) and standard deviations are reported for continuous variables with a symmetric distribution. For categorical variables, absolute (*n*) and relative frequencies are presented. The association between categorical variables was evaluated using the chi-square test or Fisher’s test, if Cochran’s rules were not met. Statistically significant associations were further analyzed by calculating standardized residuals (ri), with statistical significance indicated when ri was higher than 1.96 or lower than −1.96, assuming that the residuals follow a normal distribution. To describe the results of the CP knowledge test, a scoring system was used, with each correct response earning one point and incorrect or unanswered questions earning no points. The total score reflected the number of correct responses. This score was then included as a dependent/response variable in a linear regression, where the effect of some independent variables on the score obtained in the palliative care knowledge test was estimated, namely training, which divided the sample into groups without training, one month after training, and six months after training. The effect was evaluated by calculating the unstandardized coefficients (β) with the respective standard errors (SEs), which allowed for calculation of the *p*-value using the WALD method to decide whether to reject or not reject the null hypothesis. The Omnibus test was also performed to compare the variable-adjusted model with the null model (without the inclusion of any independent variable). The significance level considered to reject the null hypothesis was 5%.

### 2.5. Ethical Statements

This study was approved by the Local Ethics Committee (reference CE/2021/34), and the principles outlined in the Declaration of Helsinki were adhered to. Additionally, compliance with the European General Data Protection Act was ensured.

## 3. Results

The data presented refer to responses collected at three different time points, and it is possible that some participants responded during more than one phase. Therefore, the number of responses does not necessarily reflect the number of unique individuals.

A total of 93 responses were collected before training (45.58%), which correspond to Group A, 80 responses immediately after training (39.21%), which correspond to Group B, and 36 responses nearly six months after training (17.64%), which correspond to Group C.

Overall, the majority of respondents were female, with 87.1% in Group A, 83.8% in Group B, and 86.1% in Group C, with no statistically significant differences in gender distribution among the groups.

The most prevalent age group was 25 to 30 years old in Groups A (48.4%) and C (47.2%) and 31 to 40 years old in Group B (42.5%), with no differences between groups (*p* = 0.355).

Regarding workplace, the northern region of Portugal was the most represented region in all groups (37.6% in Group A, 41.3% in Group B, and 50% in Group C), with no differences among the groups (*p* = 0.654). The second most represented region was Lisbon and Vale do Tejo (LVT), followed by the central region.

The groups were statistically equal (*p* > 0.05) regarding years of experience, gender distribution, and age. Therefore, no statistically significant bias related to these characteristics was detected amongst responders.

Most doctors had less than 5 years of experience, mainly in Group A (62.4%), but also in Groups B (46.3%) and C (41.7%), with no statistically significant differences (*p* = 0.143). Regarding workplace, over 73% worked in ambulatory health centers across all groups, and there were no notable differences between the groups (*p* = 0.900).

No statistically significant link was found between having a Palliative Care Unit or Team (PCUT) at the workplace and training (*p* = 0.102), although the percentage of doctors with a PCUT at their workplace was higher in Group C (63.9%) compared to Group A (50.5%) and Group B (41.3%). Very few doctors belonged to any PCUT (less than 7% in all groups), with no differences between groups (*p* = 0.848). Interest in joining a PCUT was over 58% across all groups, with no significant differences in proportion (*p* = 0.782).

Having prior specific training in PC was associated with Group C (58.3%, ri = 2.2) compared to Group A (20.4%, ri = −2.3) (*p* < 0.001).

Professional experience in PC was not associated with the groups under study (*p* = 0.070), with 19.6% of doctors overall having professional experience in palliative medicine (PM); a higher percentage was observed in Group C (33.3%).

Self-perception of PC knowledge was significantly associated with the groups under study (*p* < 0.001), with inadequate self-perception being more prevalent in Group A (77.4%, ri = 2.8).

Table 1 presents the main professional characteristics of the sample population.

In the Appendix A, we present the results of the final knowledge assessment test from the three groups. Overall, the proportion of correct answers was high and was consistently greater than 50%, except for the question “Delirium does not normally affect memory”, where the proportion of correct answers was only 32.1%.

Group A scored an average of 22.9 points, with a median of 23 out of 27. Group B scored an average of 23.3 points, with a median of 24, and Group C scored an average of 22.8, with a median of 22.

The group of doctors surveyed one month after training scored, on average, 0.55 points higher than the doctors surveyed before training (*p* = 0.014). Male doctors scored, on average, 0.60 points less than female doctors (*p* = 0.043). Having prior specific training in PC was associated with lower scores on the PC knowledge test (*p* = 0.049). The Omnibus general significance test was statistically significant (*p* = 0.008), suggesting that the statistical model presented a better quality of adjustment than the null model.

There were no significant differences between doctors with prior professional experience in palliative medicine and those without experience in this area.

Finally, Table 2 shows the results of the relationship between behaviors and attitudes regarding PC and training. Low or very low confidence in referring a patient to a PC unit or team was higher in Group A (38.7%, ri = 2.3), reducing in Groups B (17.5%) and C (13.9%) (*p* = 0.009).

Throughout the study, there was an increase in medium, high, or very high levels of confidence (*p* = 0.005) in medicating a patient with pain who may be in their last months of life. Consequently, low and very low levels of confidence in this question decreased throughout the study: in Group A, they were 40.9%, reducing to 22.5% in Group B and particularly to 11.1% in Group C.

## 4. Discussion

Faced with a population that has a growing need for PC support [1], it is essential to understand how doctors’ knowledge in this area of medicine is acquired and developed. It is known that during pre-graduate education and medical internships, there is a lack of training for doctors [8,9,10,12,13], which raises questions about the impact of specialized training in PC following pre-graduate education.

To properly approach patients in need of PC, a particular set of skills is essential, with some shared by all health professionals and technicians, and others specific to the medical field [21,22,23]. Communication (namely, communicating “bad news”) [22], working in a multidisciplinary team [24,25,26], and supporting family members [27,28] are some of the essential skills common to all health professionals in PC. However, there are important topics that are exclusive to doctors concerning PC. The pharmacological treatment of common symptoms in PC patients [29,30,31], referral criteria [21], the management of patients in need of PC, and the adaptation and deprescription of drugs [29,32,33] are some of the topics that the entire medical profession must master.

In this study, all participants were doctors from different specialties with varying levels of expertise. It can be seen that the percentage of correct answers to questions about common topics (General Principles of CP and Communication) was much higher compared to questions about topics specific to doctors (approach to Gastrointestinal Symptoms and Neurological Symptoms). Therefore, in the future, it may be relevant to offer a more comprehensive post-graduate training program exclusively for doctors, allowing for in-depth content. However, no comparison studies have been found that consider or evaluate the usefulness of this hypothesis so far.

Although there are no published studies that have assessed knowledge before and after training conducted in the medical profession, Oliveira et al. (2021) [6] performed similar research involving medical students, where a questionnaire on PC knowledge was administered to students regardless of whether they completed a PC course during their pre-graduate training. Better theoretical knowledge was observed in students who completed the curricular unit, which aligns with the results of the present study, where doctors showed a significant improvement in their knowledge after training.

Based on the results obtained, we observed an improvement in doctors’ theoretical knowledge after training. This supports the conclusions of Oliveira (2021) [6] regarding the need for an increase/improvement in pre-graduate training for doctors in the PC area.

However, these results were not sustained six months after training, which could indicate the need for a longer training period to improve the assimilation and retention of knowledge and skills, leading to better outcomes. This is in line with the observations made by Peh (2017) [12], who verified improvements in PC knowledge after an internship for six months compared to an internship for one month.

In the present study, male participants showed worse results compared to female participants. There are conflicting findings regarding the role of gender in doctors’ knowledge in PC. Oliveira et al. (2021) [6] found greater confidence in communicating “bad news” in males, but Orlander et al. (2002) [34] found no differences between genders. Therefore, it was not possible to draw definitive conclusions regarding this association.

Contrary to the research conducted by Ghisleni et al. (2023) [15], participants with previous training in PC scored lower on the test than others. This result can be explained by the different degrees and objectives of training. Programs and teachers may vary depending on the trainees present, which may involve delving into non-specific topics, but little individualized information is provided for doctors. Based on our analysis of higher education opportunities in this area, we found that all post-graduate courses are open to various professionals, with very different basic training (doctors, nurses, social workers, psychologists, nutritionists, and physiotherapists, among others). Therefore, prior training may not be directly related to good test results.

Another important limitation comes from the demographic shift in the cohort across the three time points. Specifically, the proportion of participants with previous training or professional experience in palliative care was substantially higher in the six-month group compared to the beginning. This change introduces a potential confounding factor, as it becomes unclear whether the observed differences in outcomes were due to training effects or the presence of a more experienced and better-trained subgroup at follow-up. Although this is a limitation that cannot be corrected retroactively, it deserves careful consideration when interpreting the results. To clarify further, it would be helpful to include an additional table comparing responses over the three time points based on whether participants had previous training or experience in palliative care. This would help determine whether the intervention had a more pronounced effect on those without prior exposure to the subject.

It was observed that the independent variable “professional experience in palliative medicine” was not associated with a statistically significant increase in results compared to participants with no previous professional experience. To better interpret these results, the level of professional experience should be more precisely characterized, as doctors who have completed a short-term internship in a PC team or those working in a Continuing Care Unit (CCU) should not be considered. This can also be justified by the discrepancies that sometimes exist between theory and day-to-day, individualized clinical practice in medicine [35,36,37,38,39].

This study collected data at the following three points: before training, immediately after training, and six months later, which provides valuable longitudinal insight, especially since most similar studies focus only on immediate post-training assessments. However, the six-month follow-up group had a notably low response rate, a common limitation in long-term follow-up studies. Therefore, caution is advised when interpreting trends over time, particularly regarding sustained changes, as the smaller sample size at six months may not accurately represent the initial cohort. The results may have more qualitative than quantitative implications as a whole, giving more information about understanding and less information on generalizability, with limited conclusions for each participant and group. Furthermore, the format of the questionnaire—containing only binary Agree/Disagree questions—may have limited the depth of information gathered. Future versions could include open-ended questions to gain more detailed insights into behavioral changes. For example, participants could be asked to describe three actions they plan to implement in their clinical practice as a result of training. These responses could then be followed up at one and six months to determine if the planned changes were implemented, providing a more comprehensive assessment of the intervention’s practical effects.

An improvement was noticeable in the doctors’ increased self-confidence in referring patients for PC and in “medicating patients with pain who may be in their last months of life”. However, it can be observed that despite doctors feeling more confident in making referrals, they did not report a significant rise in the number of referrals for PC (*p* = 0.676).

## 5. Study Limitations

Like many other research studies, this one faced some limitations. Since the questionnaire was online and anonymous, it is not possible to confirm the accuracy of the answers. The questionnaire was administered at three different points, but although all participants were enrolled in the training, it was not possible to ensure that everyone answered at all three points.

The voluntary nature of enrollment in the palliative care course suggests a pre-existing interest in palliative care and motivation among the participants, which may limit the generalizability of the results to healthcare professionals who are less engaged with end-of-life care.

We can confirm that all healthcare services in which our participants were working had access to a specialized palliative care team. We did not, however, collect data on the size, structure, or specific activities of those teams, and we recognize that this limits our ability to fully interpret responses related to consultations and referrals.

The decreasing response rates over time, particularly the relatively low response rate in Group C, raises the possibility of selection bias. Participants who chose to respond at later time points may differ systematically from those who did not, potentially affecting the generalizability of our findings. In future studies with identical samples and methodologies, questionnaires administered beyond 6 months after training should only be applied to participants who responded before training.

While participation in the course was mandatory, participation in the questionnaire was optional, leading to a drop-off in the number of respondents, reflecting the time constrains of physicians regarding this topic.

We acknowledge that the lack of data from non-responders introduces a potential source of bias. Because we do not have information about their prior training in palliative care, we cannot definitively rule out the possibility that differences in the prior training levels of responders and non-responders may have influenced our results.

Although most doctors were located in a primary healthcare center, 17% of the Group A doctors said that they did not have access to a palliative structure. Among the responders one month after the survey, that number was reduced to 7%. Although the observed distribution of variables relating to training status does not meet the criteria for assuming they are statistically significant variables of influence (*p* > 0.05), the shift in the proportion of individuals with prior specific training in PC from Group A to Group C raises concern about selection bias.

Furthermore, there may be a Hawthorne effect bias, as the study participants, upon knowing the methodology and objectives from the outset, may have been able to project their knowledge and skills in line with what was expected in the study’s objectives.

It is known that social desirability bias—a tendency in survey instruments where participants answer questions in a manner that will be viewed favorably by others—may occur. The questionnaire was anonymous to minimize these effects, but it is possible that some participants may have overestimated their confidence or changed their behavior due to perceived expectations.

## 6. Conclusions

This study shows that basic training improves doctors’ palliative care knowledge, especially with longer or practical programs. About 35 h of theoretical training increased knowledge initially, but gains faded after six months. Future programs should consider longer durations, practical elements, or targeting weak areas. Besides improving knowledge, the program aimed to boost doctors’ confidence in referring patients to PC teams or units and medicating those in their last months of life, which could enhance patient care. More studies are needed to confirm this. To provide the best care, doctors must master all learning aspects.

These observations support the literature regarding the benefits of improved PC training in Portuguese medical courses. The need to establish a medical specialty in PC has also been discussed, which could strengthen palliative care development.

This study concludes that doctors can ‘know’ through training and ‘know how to do” via perception. Therefore, mastering ‘doing’ is essential for better management and medication of patients.

While the groups are, in broad terms, equivalent in relation to baseline variables like age and gender, the change in the balance of experience amongst the respondents raises doubt about the applicability of subsequent comparisons. Further work with larger cohorts is needed to address this potential.

Given the divergence between doctors’ self-confidence and their actions, practical training should be integrated. Future research could compare the impact of theoretical versus combined training on knowledge and attitudes. Since doctors’ knowledge wanes after six months, understanding trainees’ opinions could improve long-term theoretical learning. Also, more prospective and analytical instruments are necessary to access actual performance, with less self-reports. Post-graduate training should consider ongoing education platforms, like those used for chronic disease management.

## Figures and Tables

**Table 1 medsci-13-00167-t001:** Professional characterization of the 3 groups of medical doctors that attend palliative medicine course and answer study questionnaire.

	(A) Before Training (*n* = 93)	(B) 1 Month After Training (*n* = 80)	(C) 6 Months After Training (*n* = 36)	*p*
Years of professional experience				0.143 (b)
<5 years	58 (62.4%)	37 (46.3%)	15 (41.7%)	
5–10 years	18 (19.4%)	21 (26.3%)	14 (38.9%)	
11–15 years	3 (3.2%)	7 (8.8%)	1 (2.8%)	
16–20 years	4 (4.3%)	4 (5.0%)	2 (5.6%)	
21–25 years	3 (3.2%)	2 (2.5%)	3 (8.3%)	
>25 years	7 (7.5%)	9 (11.3%)	1 (2.8%)	
Specific training in palliative medicine				<0.001 (a)
No	74 (79.6%)	47 (58.8%)	16 (44.4%)	
Yes	19 (20.4%) ri = −2.3	33 (41.3%)	20 (55.6%) ri = 2.2	
Professional experience in palliative medicine				0.070 (a)
No	79 (84.9%)	65 (81.3%)	24 (66.7%)	
Yes	14 (15.1%)	15 (18.8%)	12 (33.3%)	
How do you rate your PC knowledge?				<0.001 (b)
Insufficient	72 (77.4%) ri = 2.8	33 (41.3%)	12 (33.3%)	
Satisfactory	19 (20.4%) ri = −2.5	37 (46.3%)	20 (55.6%)	
Good	1 (1.1%) ri = −2.2	10 (12.5%)	4 (11.1%)	
Great	1 (1.1%)	0 (0.0%)	0 (0.0%)	

Legend: PC: palliative medicine; *n*: number of answers; %, percentage; *p*, significance; (a) chi-square test; (b) Fisher test; standardized residuals are presented that indicate the presence of statistically significant differences when comparing observed values with expected values (ri > 1.96 or ri < −1.96).

**Table 2 medsci-13-00167-t002:** Behaviors and attitudes about palliative care and association with training.

	(A) Before Training	(B) 1 Month After Training	(C) 6 Months After Training	*p*
How do you characterize the role of the doctor in a PC Unit/Team?				0.403 (a)
Little or somewhat relevant	2 (2.2%)	4 (5.0%)	0 (0.0%)	
Very or extremely relevant	91 (97.8%)	76 (95.0%)	36 (100.0%)	
In the last 6 months, how many times have you asked opinion to a PC Unit/Team?				0.907 (a)
None or few times	72 (77.4%)	60 (75.0%)	28 (77.8%)	
Sometimes or often	21 (22.6%)	20 (25.0%)	8 (22.2%)	
In the last 6 months or so, how many patients did you refer to a PC Unit/Team?				0.676 (a)
<5	76 (81.7%)	69 (86.3%)	31 (86.1%)	
≥5	17 (18.3%)	11 (13.8%)	5 (13.9%)	
What is your level of confidence in referencing a patient to a PC Unit/Team?				0.009 (a)
Very low or low	36 (38.7%) ri = 2.3	14 (17.5%)	5 (13.9%)	
Average	21 (22.6%)	24 (30.0%)	12 (33.3%)	
High or very high	36 (38.7%)	42 (52.5%)	19 (52.8%)	
What is your level of confidence in medicating a patient with pain who may be in their last months of life?				0.005 (a)
Very low or low	38 (40.9%) ri = 2.2	18 (22.5%)	4 (11.1%) ri = −2.0	
Average	37 (39.8%)	35 (43.8%)	20 (55.6%)	
High or very high	18 (19.4%)	27 (33.8%)	12 (33.3%)	
How important is psychological support to caregivers of patients in PC?				0.517 (b)
Not important or little important	6 (6.5%)	2 (2.5%)	2 (5.6%)	
Very important	87 (93.5%)	78 (97.5%)	34 (94.4%)	
In the last 6 months, how many times have you referenced caregivers of patients in PC for psychological support?				0.480 (a)
Never or rarely	77 (82.8%)	66 (82.5%)	28 (77.8%)	
Sometimes or often	16 (17.2%)	14 (17.5%)	8 (22.2%)	

Legend: PC: palliative medicine; *n*: number of answers; %, percentage; *p*, significance; (a) chi-square test; (b) Fisher test; standardized residuals are presented that indicate the presence of statistically significant differences when comparing observed values with expected values (ri > 1.96 or ri < −1.96).

## Data Availability

Additional data could be sent upon reasonable request.

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
