# Peer review of "Is Basic Training in Palliative Care Sufficient to Guarantee the Improvement of Knowledge and Skills in This Area?—A Medical Knowledge Assessment Study"

_medsci, 2025, doi:10.3390/medsci13030167_

Round 1

Reviewer 1 Report

Comments and Suggestions for Authors

Knowledge and skills of palliative care are essential for healthcare workers managing end-of-life patients. Providing relevant trainings is necessary to enhance their confidence and improve the quality of care delivered. While the clinical question addressed is important, there are certain limitations in the study cause concerns about the representativeness and generalizability of the study findings:

(1) The response rates were 45.58%, 39.21%, and 17.64% for A, B, and C Groups, which were quite low. There was no information about non-responders. The concern is the representativeness of the study population. 

(2) In addition, it was stated "the number of responses does not necessarily reflect the number of unique individuals", which causes another concern about the comparisons between groups B and C to A that might not be able to detect the true improvement due to the results were from 3 potentially different groups. The observed improvements may be confounded by other factors, such as difference in prior PC training experience, work environment, and professional experience in PM among the three groups. 

Author Response

Reviewer 1

  • The response rates were 45.58%, 39.21%, and 17.64% for A, B, and C Groups, which were quite low. There was no information about non-responders. The concern is the representativeness of the study population. 

Answer:

We appreciate the reviewer's concern regarding the response rates for Groups A, B, and C (45.58%, 39.21%, and 17.64%, respectively), and we acknowledge that these are relatively low. However, we performed an analysis comparing key demographic and professional characteristics (age range, gender distribution, region of practice, years of experience) between the initial group of course participants and those who responded to the questionnaire at each time point (Groups A, B, and C). This analysis revealed no statistically significant differences between responders and the initial cohort on these characteristics. In the manuscript, we added this information in the results section: "The groups where statistically equal ( p > 0.05 ) on the number of years of experience, gender distribution and age. Therefore, no statistically significant bias was detected amongst responders".

We understand the reviewer's point that low response rates raise concerns about the representativeness of our study population, and we address these concerns as follows:

- Acknowledging the limitation in the manuscript: we have added a section to the limitations section of the manuscript to explicitly acknowledge this as a potential source of bias: "The decreasing response rates over time, particularly the relatively low response rate in Group C, raises the possibility of selection bias. Participants who chose to respond at later time points may differ systematically from those who did not, potentially affecting the generalizability of our findings."

- Reasons for low response rates: we believe the declining response rates are attributable to factors inherent in longitudinal online surveys involving busy medical professionals. Participation was voluntary, and the questionnaire was administered at three time points over several months. The decreasing number of respondents likely reflects a combination of factors such as time constraints, survey fatigue, and changes in professional commitments among participants. We included a sentence in limitations section: "While participation in the course was mandatory, participation in the questionnaires was optional, leading to drop-off in the numbers of respondents, reflecting time constrains of physicians on this topic".

- Addressing potential for bias through anonymity and inclusion of background data: to understand the potential differences between those that replied in the 3 phases, we analyzed the background data.

We hope this more detailed explanation addresses the reviewer's concerns. We believe the insights gleaned from this study provide valuable preliminary data, which further studies may analyze more in depth, since those are important to understand long term value of palliative basic training.

  • In addition, it was stated "the number of responses does not necessarily reflect the number of unique individuals", which causes another concern about the comparisons between groups B and C to A that might not be able to detect the true improvement due to the results were from 3 potentially different groups. The observed improvements may be confounded by other factors, such as difference in prior PC training experience, work environment, and professional experience in PM among the three groups. 

Answer:

We appreciate the reviewer's concerns about whether the potential change of individual amongst sub-groups may influence the results, and we address that the same individual is not necessary equal amongst the sub-groups. As we mentioned, this is a limitation, and was added to the main text, in the limitations section.

To address this, we performed the follow analyses:

- Addressing potential confounding by covariates: we assessed and mentioned before that age, number of years of experience, and gender were statistically equivalent. We recognize, as reviewer said, there is previous PC training. We could not understand the prior knowledge or the influence from the non responders, limiting the long term comprehension, and it will be added to limitations section: "We acknowledge that the lack of data from non-responders introduces a potential source of bias. Because we do not have information about their prior training in palliative care, we cannot definitively rule out the possibility that differences in the prior training levels of responders and non-responders may have influenced our results."

Regarding our results, although there were some results regarding previous training and PC (as shown in Table 1), although the baseline values are slightly different, this do not affect the differences in the groups.

We hope this more detailed explanation addresses the reviewer's concerns. We believe the insights gleaned from this study provide valuable preliminary data, which further studies may analyze more in depth, since those are important to understand long term value of palliative training"

- Regarding the work enviroment: this was indeed a problem, therefore we will add it to limitations session: “Although most doctors were located on a primary health care center, 17% of the A doctors said that did not had access to a palliative structure. In the responders one month after the survey, that number was reduced to 7%. Although the observed distribution of variables relating to training status does not meet criteria for assuming they are statistically significant variables of influence (p > 0.05), the shift in the proportion of individuals with prior specific training in PC from Group A to Group C raises concern for selection bias”. While the groups are in broad terms equivalent with relation to baseline variables like age and gender, the change in the balance of experience amongst the respondents raises doubt about the applicability of subsequent comparisons. Further work is needed in larger cohorts to address this potential, and we have addressed the limitation accordingly in the conclusion.

Reviewer 2 Report

Comments and Suggestions for Authors

Dear authors,

I have carefully reviewed the manuscript.

The manuscript presents an observational study based on a survey that explores improvements in Palliative Care knowledge and self-reported confidence in providing care and referring patients to specialized teams, following a basic online course for physicians.

The title, abstract, introduction, methodology, tables, and results are all clearly and appropriately written.

Comments:

a. Selection Bias and Representativeness:

The study sample is relatively small and consists of professionals who voluntarily enrolled in the Palliative Care course, suggesting a pre-existing interest and motivation in the subject. As a result, the findings from the initial survey—conducted prior to the course—reflect the views of a self-selected group and may not be generalizable to the broader population of healthcare professionals, many of whom did not express interest in the course or may be less engaged with end-of-life care.

Furthermore, the low response rates—46% before the course, 39% immediately after, and 18% six months later—raise concerns about representativeness and introduce the possibility of nonresponse bias. While the qualitative insights provided by this limited group of respondents are valuable, it would be important to expand the discussion of these limitations in the manuscript.

b. Social Desirability Bias:

Given the potential selection bias, it is worth considering whether respondents may have reported subjective improvements in confidence that align with perceived expectations after completing the course, rather than reflecting actual changes in clinical practice.

c. Contextual Bias:

The manuscript does not report whether participants had access to specialized Palliative Care teams in their workplace. This is a relevant omission, as responses regarding consultations and referrals may be influenced by the availability of such resources. The responses could only be considered representative if all participants worked in healthcare settings with access to Palliative Care services. Without this contextual information, it is difficult to assess whether the reported practices accurately reflect broader clinical realities.

In summary, while the qualitative findings are valuable, it is essential to expand the discussion of inherent biases—such as selection bias, social desirability bias, and contextual limitations—commonly associated with survey-based research, particularly in the study’s limitations section.

Author Response

Replies to the reviewers

We thank all reviewers for their comments and suggestions. Their suggestions improved the quality of the manuscript.

Please find our replies below.

Reviewer 2

  1. Selection Bias and Representativeness:

The study sample is relatively small and consists of professionals who voluntarily enrolled in the Palliative Care course, suggesting a pre-existing interest and motivation in the subject. As a result, the findings from the initial survey—conducted prior to the course—reflect the views of a self-selected group and may not be generalizable to the broader population of healthcare professionals, many of whom did not express interest in the course or may be less engaged with end-of-life care.

Furthermore, the low response rates—46% before the course, 39% immediately after, and 18% six months later—raise concerns about representativeness and introduce the possibility of nonresponse bias. While the qualitative insights provided by this limited group of respondents are valuable, it would be important to expand the discussion of these limitations in the manuscript.

Answer:

We thank the reviewer for highlighting the limitations regarding the study sample and response rates. We recognize the potential for both selection bias and non-response bias to influence our findings.

- Addressing selection bias (voluntary enrollment): The reviewer is correct that our sample consists of professionals who voluntarily enrolled in the Palliative Care course, suggesting a pre-existing interest and motivation in the subject. We acknowledge that this pre-existing interest limits the generalizability of our findings to the broader population of healthcare professionals who may not share the same level of engagement with end-of-life care.

    • We will add a sentence to the Limitations section in the manuscript: "The voluntary nature of enrollment in the Palliative Care course suggests a pre-existing interest and motivation in palliative care among participants, which may limit the generalizability of the results to healthcare professionals who are less engaged with end-of-life care."

- Addressing non-response bias (low response rates): The reviewer is also correct to point out the potential for non-response bias due to the low response rates at each time point. We acknowledge that this introduces uncertainty regarding the representativeness of the study sample and limits the conclusions we can draw regarding the overall effectiveness of the training for all participants. We think we could be able to identify correctly this limitation in the limitations section.

- Addressing qualitative insights (limited group): We agree with the reviewer that the qualitative insights provided by this limited group of respondents are valuable, we need to expand discussion on the limitations:

    • We included a sentence in the discussion section: "The results may have more qualitative than quantitative impacts as a whole, giving more information about understanding and less information on generalizability, giving limited conclusions for each participant and group”

We hope this expanded discussion adequately addresses the reviewer's concerns and provides a more balanced assessment of the strengths and limitations of our study. We believe these limitations are balanced by the benefits from results analysis.

  1. Social Desirability Bias:

Given the potential selection bias, it is worth considering whether respondents may have reported subjective improvements in confidence that align with perceived expectations after completing the course, rather than reflecting actual changes in clinical practice.

Answer:

We thank the reviewer for raising the important point about the potential for respondents to report subjective improvements in confidence that align with perceived expectations after completing the course, rather than reflecting actual changes in clinical practice. This is a significant concern in educational studies, particularly those relying on self-reported measures, and we have considered it carefully.

- Acknowledging the Possibility of Social Desirability Bias: this is particularly importante, we acknowledge this and included it in the limitations section: “It is known that social desirability bias may occur - tendency in survey instruments where participants answer questions in a manner that will be viewed favorably by others. The questionnaire was anonymous to minimize these effects, but it is possible that some participants may have overestimated their confidence or changed behavior due to perceived expectations.”

- Lack of direct measures of practice change: We recognize that our study did not directly assess changes in clinical practice, such as the number of referrals to palliative care or measurable improvements in patient outcomes. The lack of objective measures of practice change represents a limitation to the claims we can make about the real-world impact of the training. This will be pointed out now. “More prospective and analytical instruments are necessary to access actual performance, with less self-reports”, included in conclusion section.

  1. Contextual Bias:

The manuscript does not report whether participants had access to specialized Palliative Care teams in their workplace. This is a relevant omission, as responses regarding consultations and referrals may be influenced by the availability of such resources. The responses could only be considered representative if all participants worked in healthcare settings with access to Palliative Care services. Without this contextual information, it is difficult to assess whether the reported practices accurately reflect broader clinical realities.

Answer:

We appreciate the reviewer's point about access to specialized Palliative Care teams. We can confirm that all healthcare services in which our participants were working had access to a specialized palliative care team. We did not, however, collect data on the size, structure, or specific activities of those teams, and we recognize that this limits our ability to fully interpret responses related to consultations and referrals. Although all medical facilities have palliative care teams, that do not mean that they have full access. We understand these limitation and have included it in the manuscript.
